# Body mass and growth rates predict protein intake across animals

Stav Talal[1]*, Jon F Harrison[1], Ruth Farington[1], Jacob P Youngblood[1,2], Hector E Medina[3], Rick Overson[1], Arianne J Cease[1,4]

[1]School of Life Sciences, Arizona State University, Tempe, United States; [2]Department of Biology, Southern Oregon University, Ashland, United States; [3]Dirección de Sanidad Vegetal, Buenos Aires, Argentina; [4]School of Sustainability, Arizona State University, Tempe, United States

**Abstract** Organisms require dietary macronutrients in specific ratios to maximize performance, and variation in macronutrient requirements plays a central role in niche determination. Although it is well recognized that development and body size can have strong and predictable effects on many aspects of organismal function, we lack a predictive understanding of ontogenetic or scaling effects on macronutrient intake. We determined protein and carbohydrate intake throughout development on lab populations of locusts and compared to late instars of field populations. Self-selected protein:carbohydrate targets declined dramatically through ontogeny, due primarily to declines in mass-specific protein consumption rates which were highly correlated with declines in specific growth rates. Lab results for protein consumption rates partly matched results from field-collected locusts. However, field locusts consumed nearly double the carbohydrate, likely due to higher activity and metabolic rates. Combining our results with the available data for animals, both across species and during ontogeny, protein consumption scaled predictably and hypometrically, demonstrating a new scaling rule key for understanding nutritional ecology.

*For correspondence:
stalal@asu.edu

Competing interest: The authors declare that no competing interests exist.

## Editor's evaluation

How and why nutritional requirements change over development and differ between species are important questions with wide-ranging implications across a range of disciplines, from ecology to health. In this important study, Talal and colleagues set out to address these questions in laboratory and field experiments with grasshoppers, and with comparative analyses across different species. The laboratory experiments are convincing, and the study offers evidence of a universal shift from high protein to high carbohydrate intake during ontogeny.

## Introduction

Every animal must acquire a proper balance of macronutrients to maximize their performance (*Simpson and Raubenheimer, 2012*). For all animals, protein is the main building block for growing tissues, and lipids and carbohydrates (non-protein) are the primary energy fuels. Comparative studies show that different animal species have a wide range of unique protein:carbohydrate (and/or lipid) targets that optimize growth, survival, and reproduction, and these are often thought of as species-specific (*Behmer and Joern, 2008*; *Behmer, 2009*; *Hewson-Hughes et al., 2013*). While a few studies indicate developmental effects on macronutrient intake, we lack a clear understanding about how and why ontogeny or body size affect macronutrient consumption and intake targets (*Ojeda-Avila et al., 2003*; *Peters, 1983*; *Wang et al., 2019*). To address this lack, we determined the effect body mass throughout ontogeny on macronutrient (protein and carbohydrate) intake and growth rate for the

polyphagous and transboundary migratory pest, *Schistocerca cancellata* (Serville, 1838), the South American locust, and integrated our results with prior studies of this topic in animals.

Foraging decisions are often driven by the need to balance protein (p) with non-protein (np) energy because these macronutrients make up the vast majority of a consumer's diet and food sources rarely match the balance needed. The relative macronutrient requirements of individuals across development and the factors that influence these intake targets have profound implications for population dynamics and ecosystems, particularly for herbivores. For example, in many cases, growth and population levels of freshwater invertebrate herbivores are limited by protein (or more specifically, essential amino acid) availability (*Fink et al., 2011*). In contrast, late instars of grasshoppers and some lepidopteran caterpillars have low protein to carbohydrate targets due to their high energy requirements for adult flight (*Lee et al., 2004*; *Talal et al., 2020*). In these cases, low nitrogen environments which harbor low-protein, high-carbohydrate plants promote outbreaks and devastating locust migratory swarms (*Cease et al., 2012*). Animals restricted to feeding on foods that diverge from their required p:np balance can experience pronounced performance deficits in development time, mass, reproduction, and survival (*Behmer and Joern, 2008*; *Behmer, 2009*; *Raubenheimer et al., 2022*; *Simpson and Raubenheimer, 2012*; *Talal et al., 2020*).

The Geometric Framework for Nutrition (*Simpson and Raubenheimer, 2012*) was developed to study how organisms balance multiple nutrients, and identifying intake targets is a key principle. Most organisms will self-select a balance of nutrients, and this can be tested by giving individuals a choice between two or more foods differing in the ratio of two or more nutrients. Individuals differentially eat the diets to achieve an intake target. Macronutrient targets can vary across species. For example, cats selectively consume and perform better on more protein-biased diets (52p:48np) than dogs, for which a 30p:70np diet is optimal (*Hewson-Hughes et al., 2013*; *Hewson-Hughes et al., 2011*). Such variation in nutritional targets can occur even among closely related species, and we are beginning to understand some of these patterns. For example, late-instar juveniles of seven species of congeneric grasshoppers that share the same habitat exhibit widely different species-specific p:np targets that maximize their growth performance (*Behmer and Joern, 2008*). Domestic dogs are omnivorous while wolves are carnivorous, likely due to the availability of diverse foods domesticated dogs can obtain from humans (*Bosch et al., 2015*). Within arthropods, predators prefer food with relatively more protein than their herbivorous prey, whose food is often lower in protein concentration (*Wilder et al., 2013*).

Some evidence suggests that intake of protein relative to carbohydrate (and or lipids) may generally decrease through ontogeny. Stable isotope analysis of tooth and skin suggested that mass-specific protein consumption declines during ontogeny in bottlenose dolphins (*Tursiops truncates*, *Knoff et al., 2008*). Similarly, in turtles (*Trachemys scripta*, *Bouchard and Bjorndal, 2006*), lizards (*Stellagama stellio*, *Karameta et al., 2017*), and sturgeons (*Acipenser persicus*, *Babaei et al., 2011*), food preference, digestive efficiency, and digestive enzymatic activities indicate decreasing mass-specific protein assimilation and need as ontogeny progresses. Decreases in the ratio of p:np in milk through ontogeny also suggest that offspring nutritional requirements shift with age. In northern elephant seals (*Mirounga angustirostris*), the lipid concentration of milk increases by approximately fivefold during 30 days of lactation (*Riedman and Ortiz, 1979*), while in humans, the protein concentration of milk decreases as lactation progresses (*Ballard and Morrow, 2013*; *Bauer and Gerss, 2011*). Tammar wallabies provide milk with a lower p:np ratio to older offspring, even when nursing two offspring simultaneously (*Nicholas et al., 1997*). A few studies have tested for an effect of ontogeny or body mass on preferred p:np consumption and utilization in invertebrates, but generally only over a short span of the life cycle. A study of brown-banded cockroaches showed that the self-selected ratio of casein:glucose decreased from third to final instar (*Cohen et al., 1987*). Lepidopteran caterpillars decrease p:np consumption over three instars (*Stockhoff, 1993*). We lack studies of how macronutrient targets are affected across whole-ontogeny or across broad body size ranges of species, and these are necessary to provide an understanding of whether a decline in mass-specific protein intake is a general pattern among animals.

Macronutrient consumption can also vary in response to environment and activity levels. For example, viral-infected caterpillars shifted toward a new self-selected macronutrient ratio that maximized survival (*Cotter et al., 2011*). During winter months (cold weather), golden snub-nosed monkeys increased their daily non-protein energy intake, probably due to the increased cost of thermoregulation

(*Guo et al., 2018*). Many migratory birds adjust their nutrition to facilitate adequate fat accumulation (*Bairlein, 2002*). In early development, human energy requirements are highly correlated with basal metabolic rate and growth processes (0–6 months) (*Butte et al., 2002*). However, later when a child increases their physical activity, energy requirements correlate highly with activity level (reviewed in *Savarino et al., 2021*). For example, a single high-intensity exercise increases lipid consumption in humans (*Klausen et al., 1999*).

Based on this literature, we hypothesized that animals would steadily reduce protein consumption during ontogeny because mass-specific growth rate declines (*Brown et al., 2004*; *West et al., 2001*; *White et al., 2022*), causing a progressive decrease in the consumption of protein relative to carbo- hydrate. We tested this hypothesis using South American locusts, *S. cancellata*. We predicted that mass-specific protein consumption would decrease strongly during development, in tight correlation with a decrease in mass-specific growth rate and a decrease in the protein:carbohydrate intake ratio, and that these relationships would hold across all animals because growth rate scales hypometrically across animals of different body sizes as well as during ontogeny (*Hatton et al., 2019*; *Peters, 1983*; *West et al., 2001*). To partially test whether our lab results could predict the macronutrient require- ments in more ecological-relevant conditions, we also compared intake targets between lab-reared animals with field-collected locusts at one developmental stage.

**Table 1.** The diet pair presented did not affect the amount of protein and carbohydrate consumed at any developmental stage (multiple analysis of covariance [MANCOVA], with diet pairs as blocks and masses as a covariate), indicating that locusts tightly regulated to a specific intake target.

| Nymph instar | | F-value | p-Value | Wilks' $\Lambda$ |
|---|---|---|---|---|
| First $N_{male} = 37$ $N_{female} = 41$ | Diet | $F(2,72) = 2.82$ | 0.07 | 0.93 |
| | Sex | $F(2,72) = 1.46$ | 0.24 | 0.96 |
| | Diet × sex | $F(2,72) = 0.36$ | 0.7 | 0.99 |
| Second $N_{male} = 46$ $N_{female} = 51$ | Diet | $F(2,91) = 2.33$ | 0.1 | 0.95 |
| | Sex | $F(2,91) = 2.63$ | 0.08 | 0.95 |
| | Diet × sex | $F(2,91) = 0.10$ | 0.91 | 0.99 |
| Third $N_{male} = 50$ $N_{female} = 48$ | Diet | $F(2,93) = 1.09$ | 0.34 | 0.98 |
| | Sex | $F(2,93) = 10.1$ | <0.001 | 0.82 |
| | Diet × sex | $F(2,93) = 0.87$ | 0.42 | 0.98 |
| Fourth $N_{male} = 58$ $N_{female} = 40$ | Diet | $F(2,91) = 0.38$ | 0.68 | 0.99 |
| | Sex | $F(2,91) = 2.84$ | 0.06 | 0.94 |
| | Diet × sex | $F(2,91) = 0.32$ | 0.73 | 0.99 |
| Fifth $N_{male} = 55$ $N_{female} = 35$ | Diet | $F(2,84) = 1.17$ | 0.32 | 0.97 |
| | Sex | $F(2,84) = 2.84$ | 0.06 | 0.94 |
| | Diet × sex | $F(2,84) = 0.82$ | 0.44 | 0.98 |
| Sixth $N_{male} = 30$ $N_{female} = 24$ | Diet | $F(2,48) = 0.91$ | 0.41 | 0.96 |
| | Sex | $F(2,48) = 1.18$ | 0.32 | 0.95 |
| | Diet × sex | $F(2,48) = 1.37$ | 0.26 | 0.95 |
| Adult $N_{male} = 28$ $N_{female} = 25$ | Diet | $F(2,47) = 0.752$ | 0.477 | 0.969 |
| | Sex | $F(2,47) = 5.191$ | 0.009 | 0.819 |
| | Diet × sex | $F(2,47) = 0.184$ | 0.832 | 0.992 |

## Results

### Protein-to-carbohydrate intake ratio decreased throughout development

We measured self-selected protein and carbohydrate consumption rates for each developmental stage (instars, adults) of *S. cancellata* using chemical-defined artificial diets (see 'Materials and methods' for more information). We found that younger instars (first to fourth) had a protein-biased consumption (selected high protein-to-carbohydrate ratios, p:c) with third-instar nymphs exhibiting the highest p:c of 1.37p:1c (*Figure 1A and D*). In contrast, older locusts became carbohydrate-biased, with adults selecting intake targets of 1p:2.66c (*Figure 1A and D*). Males and females (both unmated) did not differ from each other in relative macronutrient consumption during most of the developmental stages (*Figure 1*, *Table 1*). There were no significant interactions between sex and diet pair on total macronutrient consumption (*Table 1*). The insignificant effect of diet pair indicates that locusts are tightly regulated to a specific intake target. The mortality was relatively low and was not affected by sex or diet pairs.

Mass-specific carbohydrate consumption rates were about 30% higher for the first two instars compared to older animals but varied little across the older groups (ANOVA: diet: $F_{6,554} = 34.459$; p<0.001, *Figure 1B*). Males and females did not differ significantly in mass-specific carbohydrate consumption rates (ANOVA: sex: $F_{1,554} = 0.294$; p=0.940, *Figure 1B*). Mass-specific protein consumption rate decreased steadily through ontogeny, with a roughly fourfold decrease in adults compared to first instars (ANOVA: diet: $F_{6,554} = 193.142$; p<0.001, *Figure 1C*). There were differences between the sexes (ANOVA: sex: $F_{1,554} = 7.055$; p=0.008) and a significant interactive sex * diet effect on mass-specific protein consumption (ANOVA: sex * diet: $F_{6,554} = 38.995$; p=0.011), which was associated with small, irregular stage effects on which sex consumed more. Together, these ontogenetic effects on carbohydrate and protein consumption led to strong decreases in the protein:carbohydrate intake ratio through ontogeny, with the youngest instars consuming about 30% more protein than carbohydrate and the oldest juveniles and adults consuming approximately twice as much carbohydrate as protein (ANOVA: sex: $F_{1,574} = 3.112$, p=0.078; developmental stage: $F_{6,574} = 87.529$, p<0.001; sex * developmental stage: $F_{6,574} = 1.419$, p=0.645) (*Figure 1D*).

### Macronutrient consumption correlates with growth, but only protein consumption consistently scales hypometrically

The decrease in protein consumption (corrected by initial mass) was well-predicted by the decrease in specific growth rates (see 'Materials and methods') in both sexes (*Figure 2A*). Mass-specific carbohydrate consumption (corrected by initial mass) was also negatively correlated with specific growth rates, but this was only significant for females (*Figure 2B*). Plotting macronutrient consumption rates on a log-log plot revealed a strong correlation with body mass (*Figure 3*). Whereas protein consumption rates in *S. cancellata* scaled strongly hypometrically, with a slope of 0.761 (95% confidence interval: 0.744–0.778) (*Figure 3A*), carbohydrate consumption rates scaled weakly hypometrically, with a slope closer to 1 (slope of 0.939; 95% confidence interval: 0.92–0.957) (*Figure 3B*).

Combining our data with the available literature for animals (see 'Materials and methods' for more details) revealed that declining protein consumption rates during ontogeny or across species that differ in body mass is a general pattern for animals (*Figure 3A*). Older and larger animals consume proportionally less protein in locusts, fish, rats, chickens, pigs, cats, caribou, and dairy cattle, and with a very similar pattern holding across species (*Figure 3A*). Correcting for phylogeny (see 'Materials and methods') yielded a multispecies regression model (slope of 0.776) that predicts protein consumption rates by mass, with a very similar slope to the regression for data from *S. cancellata* (*Figure 3A*).

### Field-collected nymphs had higher rates of metabolism and carbohydrate consumption but similar protein consumption as lab-reared locusts

Field-collected (Gran Chaco, Paraguay, April 2019) South American locusts had more carbohydrate-biased intake targets relative to lab-reared locusts (*Figure 1A*). Death rates were low during the experiments, and there was not a significant effect of diet on the death rate. Male fifth- and sixth-instar nymphs collected from field populations had 50–90% higher carbohydrate consumption rates

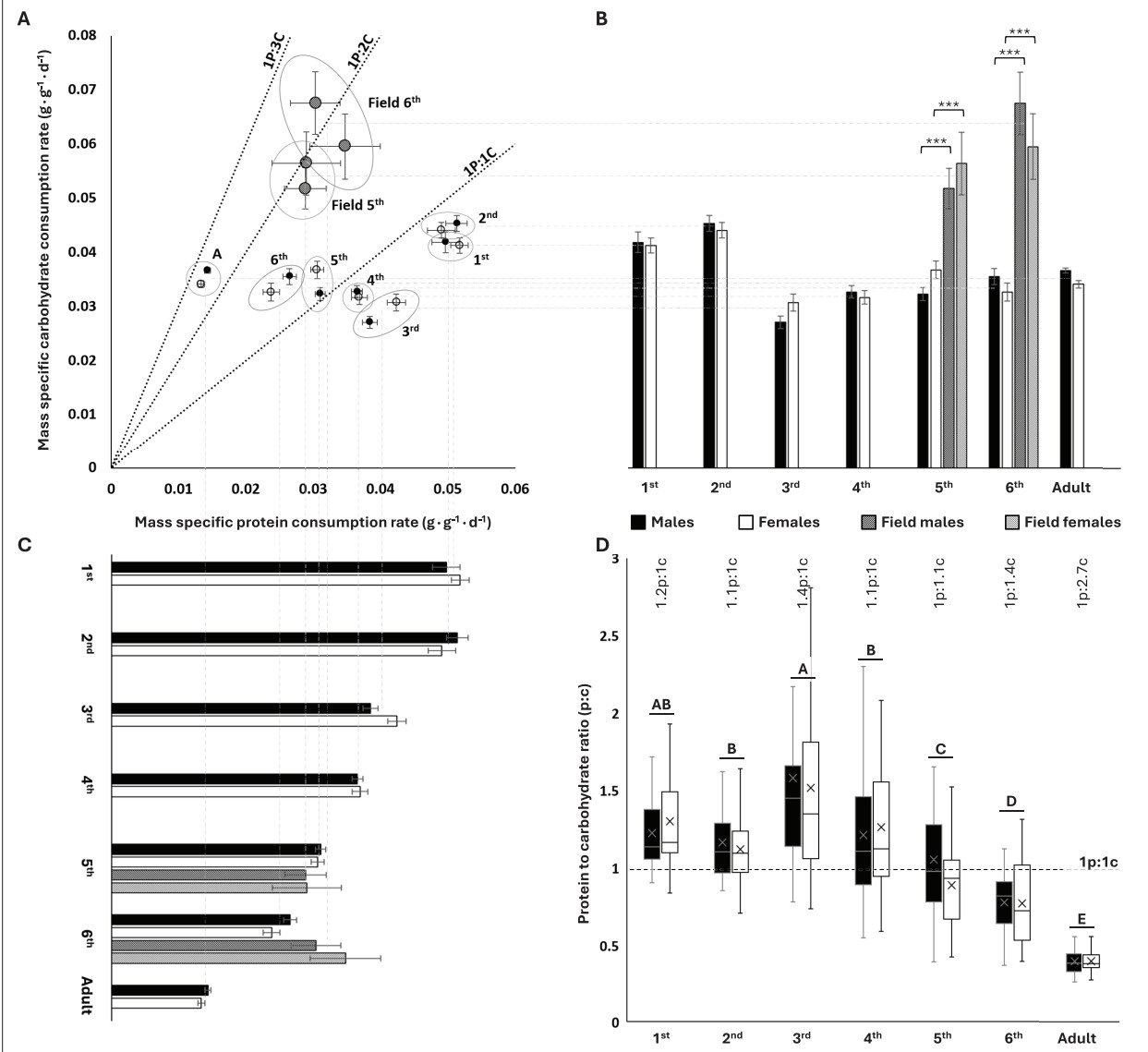

**Figure 1.** Protein and carbohydrate consumptions across different divelopment stages in lab and flield population of *Schistocerca cancellata*. (**A**) Self-selected protein to carbohydrate (p:c) consumption rates decreased systematically during ontogeny. (**B**) For lab-reared locusts, mass-specific carbohydrate consumption rates were highest in early instars relative to older instars and adults. Field-collected fifth and sixth instars consumed more carbohydrate than lab-reared nymphs. (**C**) For lab-reared locusts, mass-specific protein consumption declined systematically with age. Field-collected fifth- and sixth- instar nymphs consumed protein at similar rates to lab-reared animals. (**D**) Young, first to fourth nymph instars self-selected protein-biased intake target ratios, whereas later in development, locusts became carbohydrate-biased (medians and interquartile ranges are represented by the boxes and center line, with an X to indicate the mean). The numbers above the boxes represent life stage averaged (both sexes) p:c intake targets. The post hoc letters were given only when there was no significant interactive developmental stage * sex effect. For panels (**A–C**), means and standard errors (SEM) are shown. All consumption rates are in grams per day, divided by the final body mass of the relevant instar. The three asterisks represent significant differences between lab and field populations when p<0.001. Throughout, males are black circles/bars and females are white circles/bars; field locusts are represented by striped bars. For sample sizes, see *Table 1*.

The online version of this article includes the following source data for figure 1:

**Source data 1.** Numerical data of *Figure 1*.

**Source data 2.** Numerical data of *Figure 1* (field-collected locusts).

relative to lab-reared nymphs (Mann–Whitney *U* test: *U* = 2; *U* = 17; respectively; p<0.001 for both instars) as did female fifth- and sixth-instar nymphs (Mann–Whitney *U* test: *U* = 6; *U* = 2; respectively; p<0.001 for both instars) (*Figure 1A and B*). However, there were no significant differences in protein consumption between field-collected and lab-reared nymphs for male fifth- and sixth-instar

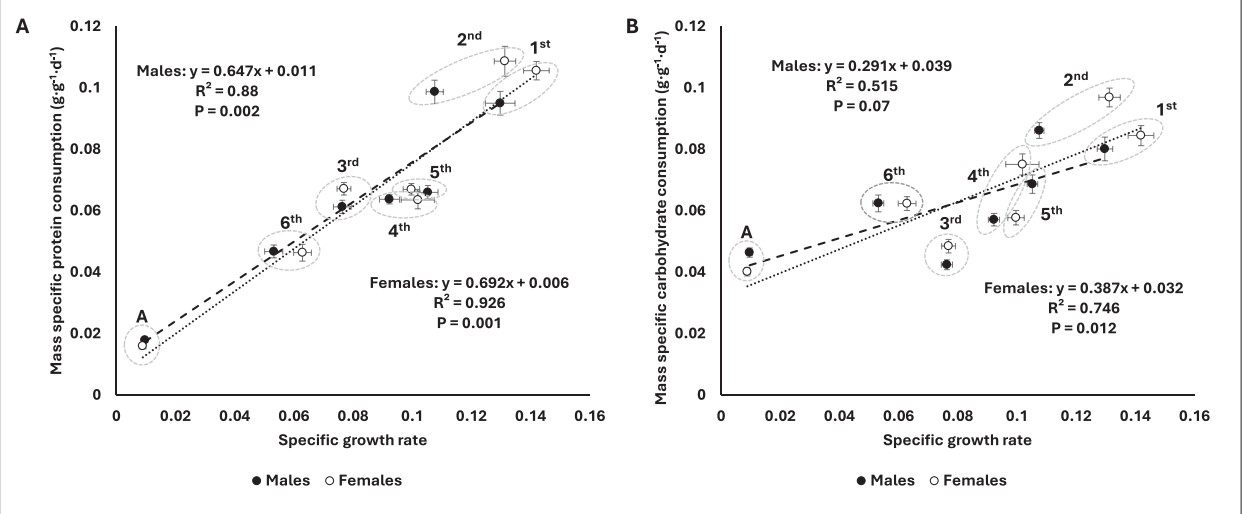

**Figure 2.** Mass-specific protein consumption rate was well-predicted by specific growth rate across ontogeny in both sexes (**A**), whereas mass-specific carbohydrate consumption was only significantly related to specific growth rate in females (**B**). Filled circles and dashed line represent males, whereas opened circles and dotted line represent females. Here, consumption rates in grams per day were divided by initial mass at the relevant instar because the final mass at the instar was a strong determinant of the parameters on both axes. Means and standard errors (SEM) are shown.

The online version of this article includes the following source data for figure 2:

**Source data 1.** Numerical data of *Figure 2*.

nymphs (Mann–Whitney $U$ test: $U = 204$; p=0.197; $U = 163$; p=0.135; respectively) or female fifth- and sixth-instar nymphs (Mann–Whitney $U$ test: $U = 43$; p=0.071; $U = 127$; p=0.859; respectively) (*Figure 1B and C*). The higher carbohydrate consumption of field-captured locusts was partly due to a higher resting metabolic rate. Using stop-flow respirometry (see 'Materials and methods'), we demonstrated that field-collected sixth- (N = 29) instar nymphs had ~23% higher mass-specific resting oxygen consumption rate than sixth- (N = 50) instar lab-reared nymphs (1.126 ± 0.052 ml·g$^{-1}$·h$^{-1}$, 0.914 ± 0.021 ml·g$^{-1}$·h$^{-1}$, mean ± SEM for field-collected and lab-reared, respectively) (Mann–Whitney $U$ test: $U = 349$; p<0.001).

## Discussion

Overall, our results demonstrate that macronutrient targets change predictably from high protein:carbohydrate consumption in the young toward increasingly lower protein:carbohydrate intake targets during ontogeny in *S. cancellata*. From first instar to adult for *S. cancellata*, mass-specific protein consumption rate decreased fourfold with little change in mass-specific carbohydrate consumption (*Figure 1*). The decrease in mass-specific protein consumption rate was tightly correlated with a decline in specific growth rate, likely explaining the shift in protein requirements (*Figure 2A*). However, intake targets measured in the lab did not well-predict intake targets in the field, as protein demand did not differ between lab and field populations, but carbohydrate consumption rate was >50% higher in field populations (*Figure 1*).

It is important to note that we have not measured the fitness consequences of variation in diet composition across the various locust instars, so we cannot claim that the observed decline in protein:-carbohydrate intake ratio is beneficial. This is a complex issue because the fitness consequences of larval diet can be measured in many ways, including growth and survival of the larvae, and adult reproduction and longevity; and these traits do not always correlate (*Sentinella et al., 2013*). However, the fact that many other studies have found that measured intake targets optimize fitness suggests that this pattern is a beneficial one (*Raubenheimer and Simpson, 2018*).

Here it is demonstrated for the first time that protein consumption rates decrease during ontogeny in a predictive way with animal mass. In our experiments with *S. cancellata*, we cannot determine the extent to which the observed pattern is due to developmental or body mass changes, though the observation that the pattern is similar to that seen across species differing in mass suggests that

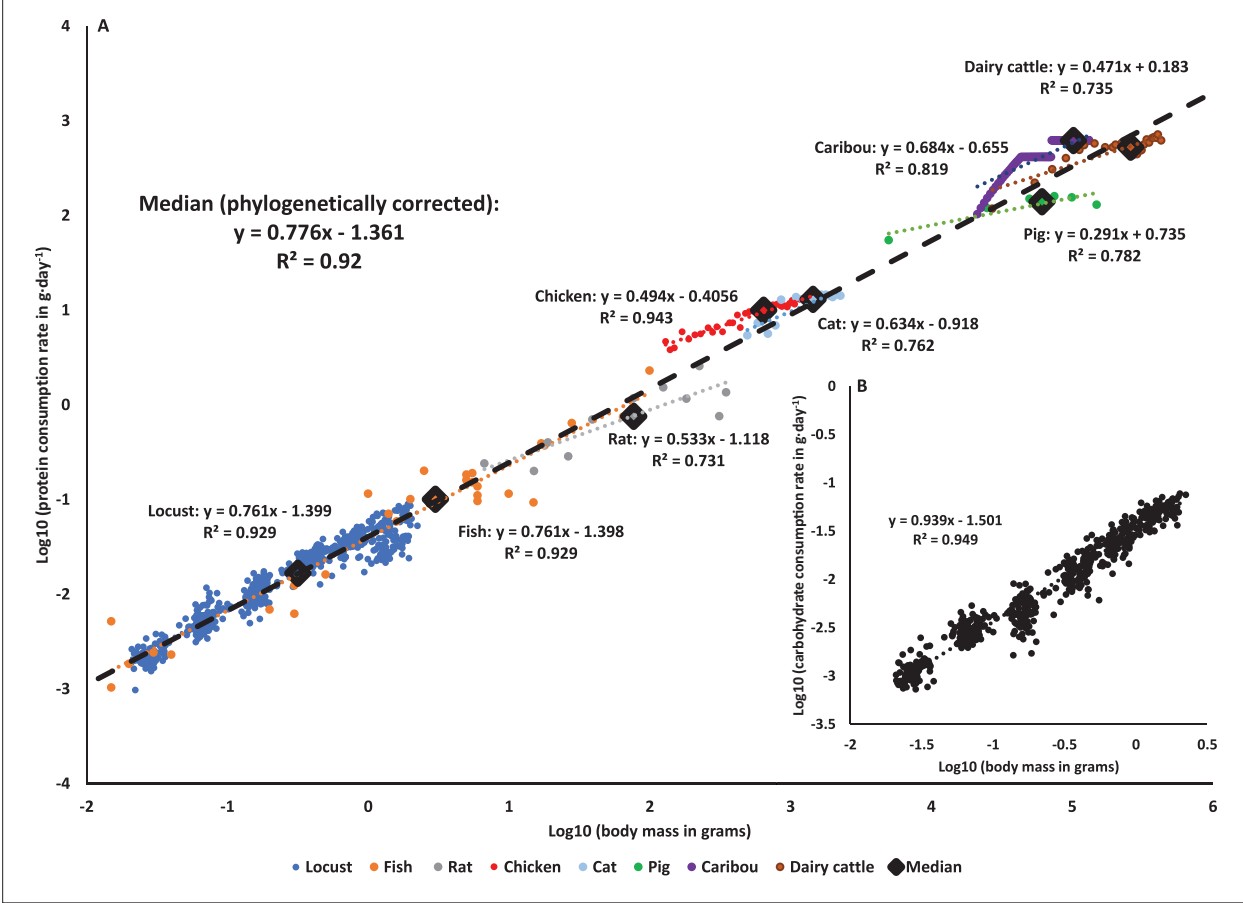

**Figure 3.** Macronutrients scaling through ontogeny across animal kingdom. (**A**) Protein consumption rate scales hypometrically throughout development across the animal kingdom (PGLS, slope = 0.776, SE = 0.086, $t$ = 9.036, p<0.001). The blue circles: locusts (*Schistocerca cancellata*, this study); orange circles: fish (early development in multiple species [reviewed in **Dabrowski, 1986**]); gray circles: rats (*Rattus rattus* [**Ricci and Levin, 2003**]); red circles: chicken (*Gallus gallus domesticus* [**Kaufman et al., 1978**]); light blue circles: cats (*Felis catus* [**Dickinson and Scott, 1956**; **Miller and Allison, 1958**]); green circles: pigs (*Sus domesticus* [**Black et al., 1986**]); purple circles: caribou (*Rangifer tarandus* [**McEwan, 1968**]); brown circles: dairy cattle (*Bos taurus* [**Crichton et al., 1959**]). Diamonds represent the median value of each taxonomic group (matched by color), and the black dashed line is the across-species phylogenetically corrected regression model (see **Figure 3—figure supplement 1** for phylogenetic tree). (**B**) Carbohydrate consumption rates scale hypometrically, a very close to isometrically in *Schistocerca cancellata*.

The online version of this article includes the following source data and figure supplement(s) for figure 3:

**Source data 1.** Numerical data of **Figure 3** (locusts).

**Source data 2.** Numerical data of **Figure 3** (animals).

**Figure supplement 1.** Phylogenetically correction and scaling of protein consumption rate across the animal kingdom.

variation in body size is responsible. Thus, protein consumption can be added to the list of traits that scale predictably with body size (**Schmidt-Nielsen, 1995**; **Sibly et al., 2012**). The hypometric scaling of protein consumption across species is consistent with the general hypometric scaling of growth rates across animals (**Hatton et al., 2019**). Though ontogenetic slopes of protein consumption on mass were much lower than the cross-species pattern in a few groups, including cats and pigs, it will be interesting to determine whether such variation relates to interspecific variation in the scaling of ontogenetic growth and lifespan.

Assuming that energy needs and consumption are primarily set by metabolic rate, we would expect that mass-specific non-protein consumption to decrease with both animal mass and age due to the generally observed hypometric scaling of metabolic rates across animal taxa (**Harrison et al., 2022**; **White et al., 2022**). In locusts, we demonstrated that carbohydrate consumption scaled hypometrically, but with a slope very close to 1, a much higher mass-scaling exponent than observed for protein consumption (**Figure 3A**), but in the range of reported scaling for resting metabolic rate (0.77–1)

for orthopterans (*Fielding and DeFoliart, 2008*; *Greenlee and Harrison, 2004*). Likely, in locusts, carbohydrate consumption of older individuals is increased due to the increase in mass-specific lipid stores that occurs in older juveniles and adults, as stored lipids are mainly synthesized from ingested carbohydrates (*Talal et al., 2021*). In addition, we demonstrated a positive correlation between mass-specific carbohydrate consumption rate and specific growth rate, with the highest of both parameters in early development (*Figure 2B*). This could be explained by the energy cost of new tissue growth and new protein synthesis, which are the highest in early development (*Li et al., 2019*; *Pace and Manahan, 2007*), and match the protein requirements during this period of time.

An important goal for the field of nutritional ecology is to predict nutritional needs, foraging behavior and strategies, and consequences of nutritional imbalance for animals in the field (*Behmer and Joern, 2008*). Relative to the lab population, we measured a 50–90% increase in carbohydrate consumption rates for field-collected fifth- and sixth-instar nymphs. In contrast, protein consumption rates did not vary between lab and field in our study. This may not be true under every ecological condition; for example, poor resource conditions that reduce growth will likely also reduce protein consumption. Nonetheless, these data support the hypothesis that protein consumption rates of animals in good field conditions may be predicted from results with lab-reared animals. There are multiple reasons why carbohydrate consumption in the field may be poorly predicted by laboratory consumption data. Consumption patterns can reflect their past feeding history (*Marmonier et al., 2000*; *Wiggins et al., 2018*), which was not known in the case of our field-captured animals. In the lab, consumption rates were measured from the first day of the instar, but likely over a later part of the instar in the field-collected animals, potentially affecting the results. Captive animals usually do not need to travel long distances to forage, which can be energetically expensive and cause long-term effects on resting metabolic rates (*Bergman et al., 2001*). Studies of monkeys and apes have demonstrated that decreases in foraging activity in captivity may promote metabolic suppression, diabetes, and obesity (reviewed in *Bellisari, 2008*). Increased energy demands and energy metabolism in field animals may also be due to a past history of consumption of tougher, better chemically defended plants (*Clissold et al., 2009*; *Maskato et al., 2014*). Field animals may be more likely to be coping with pathogens, and immune responses can elevate metabolic rates in insects (*Catalán et al., 2012*; *Freitak et al., 2003*). In addition, adaptation to lab conditions over multiple generations in captivity may reduce metabolic rates and carbohydrate consumption (*Garland et al., 1987*; *Latorre et al., 2020*). Also, it is important to note that because we only tested one instar in the field, we have not demonstrated that hypometric scaling of protein consumption occurs under field conditions, though this seems likely. Future studies will be necessary to confirm this, and to decipher the mechanisms that elevate metabolic rates and carbohydrate consumption for locusts and other animals in the field.

## Conclusions and future directions

Hypometric scaling of protein consumption is associated with declining specific growth rate during ontogeny and body mass across species in animals, providing a new and useful paradigm for nutritional ecology. Many important questions remain. Is species-level variation in the ontogenetic scaling of protein consumption rate predictable by species differences in growth rates? How useful would age-specific diets be for humans and animal husbandry? Is the hypometric scaling of protein intake related to parallel patterns in the morphology and physiology of digestive and assimilative processes? Does spatial or temporal variation in protein availability play an important role in the biogeography of animal body sizes? Plausibly, higher protein availability favors the ecological success of smaller, faster-growing animals. Finally, rising temperatures and $CO_2$ levels are predicted to lower the relative availability of protein to carbohydrate in leaves; while it has been shown that this can slow herbivore growth (*DeLucia et al., 2012*; *Kuczyk et al., 2021*; *Scherber et al., 2013*), our findings suggest such changes may also select for herbivores with larger body sizes, higher activity, and lower mass-specific protein requirements.

## Materials and methods
### Locust lab culture

We used South American locusts (*S. cancellata*) from a captive colony at Arizona State University (ASU), 7–10 generations after locusts were collected from La Rioja and Catamarca regions of Argentina. The

culture was kept at 30% RH, 34°C during the day and 25°C during the night, under 14 hr light:10 hr dark photoperiod. Supplementary radiant heat was supplied during the daytime by incandescent 60 W electric bulbs next to the cages. In this general culture, locusts were fed daily with wheat shoots, fresh romaine lettuce leaves, and wheat bran ad libitum. For all experiments, animals were excluded only if they died during the experimental procedure.

## Artificial diets

The artificial diets were made as described by *Dadd, 1961* and adapted by *Simpson and Abisgold, 1985*. We used five different isocaloric artificial foods in different assays that varied in protein and digestible carbohydrates: 7p:35c (% of protein and % of digestible carbohydrates, by dry mass), 14p:28c, 21p:21c, 28p:14c, 35p:7c. All the diets contained 54% cellulose and 4% vitamins and salts. The proteins were provided as a mix of 3:1:1 casein:peptone:albumen. The carbohydrate was provided as a 1:1 mix of sucrose and dextrin.

## Effect of ontogeny and body mass on intake targets

Nutritional intake targets were measured for each nymphal instar (50–60 individuals for each sex for first to fifth and 30 individuals for each sex for sixth), with diets weighed on the first day and last day of each instar. Animals were kept in individual cages with an air temperature of 34°C:25°C (day:night), without access to a radiant heat source. The adult (30 for each sex) intake targets measurements were started on molt day and recorded for 3 weeks. To have sufficient individuals of the same age, in each developmental stage, we monitored for newly molted individuals and randomly collected them on the same day. For the first-instar nymphs, we monitored egg cups daily. When hatching was observed, within a few hours, we inserted the cups into standard colony-rearing cages (45 × 45 × 45 cm metal mesh) to keep the ages of the nymphs as similar as possible. Sexing was performed by identifying the presence/absence of developing ovipositor valves. For early developmental stages (first to third instars), we used a dissecting microscope to visualize these structures (SMZ-168, MOTIC, Schertz, TX).

During these measurements, individuals were kept in plastic containers with holes drilled in the roof for ventilation which maintained the RH at ~30%. The first- to third-instar nymphs were kept in 11 × 16 × 4 cm cages, and fourth-instar nymphs to adults were kept in 19 × 10.4 × 14 cm containers. Each container had a water tube (refilled once a week), a perch (for successful molting) and two complementary artificial diets. To determine if locusts were arriving at a consistent p:c intake target ratio and not just eating randomly from the two dishes, we provided half the locusts with the choice between 35p:7c and 7p:35c diets, while the other half were provided with the choice between 28p:14c and 7p:35c diets. We randomly placed cages from different diets pair treatments and sex on different shelves. To calculate consumption, we weighed each diet dish two times: (1) after drying and prior to inserting it into the assay boxes and (2) after it was removed from the experimental boxes and re-dried at 60°C for 24 hr. To reduce error during the first-instar nymph experiment, we used small diet dishes (made from 1.5 ml Eppendorf lids) and weighed them with a microbalance (MSA6.6S-000-DM, accuracy of $10^{-6}$ g, Sartorius Weighing Technology GmbH, Goettingen, Germany). For all other instars, we used diet dishes made from an acrylic cylinder (10 × 25 mm) glued to a Petri dish (58 mm in diameter) and weighed them using an analytical balance (accuracy of $10^{-5}$ g, XSE205, Mettler Toledo, Columbus, OH). Locusts were also weighed using the analytical balance. To reduce handling, which increases mortality, we weighed the locusts only after the experiment and used final masses to correct consumption values. To calculate specific growth rate (*Equation 1*) for each instar, we calculated mean initial masses for an extra 20 freshly molted (or newly hatched for first instar) individuals for each instar and sex.

$$Specific\ growth\ rate = \frac{\ln\left(\frac{final\ mass}{initial\ mass}\right)}{instar\ developmental\ time\ (days)} \tag{1}$$

For comparisons of consumption rate to specific growth rate, we divided consumption rates in grams per day by initial mass at the relevant instar because final (but not initial) mass was a strong determinant of specific growth rate.

## Comparing intake targets and metabolic rates between lab-reared and field-captured locusts

We compared protein and carbohydrate consumption rates of fifth- and sixth-instar lab-reared nymphs (from the intake target experiment) to the field data we randomly collected from similarly aged nymphs in 2019 during the *S. cancellata* outbreak in Gran-Chaco, Paraguay (*Talal et al., 2020*). During the days of the experiments, field-collected nymphs were kept at temperatures averaging 32.2 ± 1.94°C (measured with a Hobo logger, Onset, Bourne, MA), without access to a radiant heat source. We assessed macronutrient consumption rates by providing locusts with a choice between a low and a high carbohydrate diet (the same diets as in intake target experiment, see above) for 8 days.

Comparison of resting metabolic rates was carried out on sixth-instar nymphs that were reared on confined artificial diets varying in protein:carbohydrate ratio (both in the field and in the lab) (*Talal et al., 2021*). Since we did not find an effect of dietary protein to carbohydrate ratio on oxygen consumption in either the lab or in the field (*Talal et al., 2021*), we pooled the data from the different diet treatment groups to compare resting metabolic rates between lab and field populations by measuring oxygen consumption. We carried out stop-flow respirometry using a FoxBox oxygen analyzer (Sable Systems International, Las Vegas, USA) as described in *Talal et al., 2021*. Briefly, after inserting the nymph in a metabolic chamber and flushing it with $CO_2$-free, dry, air, the chamber was sealed for a period of time, after which a known volume was injected into $CO_2$-free, dry air flow (500 ml·min$^{-1}$) which was flushed through the oxygen analyzer. The metabolic rate (oxygen consumption) was temperature-corrected to 34°C using $Q_{10}$ of 2 (*Talal et al., 2021*).

## Scaling of protein consumption across animals

To determine whether the pattern of protein consumption scaled similarly across animals as in locusts, we survey the literature for measures of protein consumption relative to body mass in animals during ontogeny. We searched the literature using scholar.google.com using the search terms 'protein require-ment during development/ontogeny; macronutrient consumptions/requirements; self-selection of protein consumption during development'. We included any study that measured body masses and protein consumption rates over ontogeny, as well as studies with these data for adults. Because temperature has strong effects on metabolic rates and consumption rates, we corrected the data for ectotherms to 37°C using a $Q_{10}$ of 2 (*Clarke and Johnston, 1999*; *Harrison and Fewell, 1995*).

## Statistics

Statistical analyses were performed using SPSS 20.0 (IBM) and R Studio (*R Development Core Team, 2021*). Prior to using parametric analyses, the normality of data was confirmed. For the intake target experiments: to rule out random feeding on different diet pairs, we employed multiple analysis of covariance (MANCOVA), using mass of carbohydrate and protein eaten as dependent variables, diet pair and sex as independent variables, and final body mass as a covariate. Due to some assumption violations, we compared protein and carbohydrate consumption rates as well as p:c ratios, among developmental stages and sexes, using aligned rank-transformed observations on mass-specific values. To test for a significant effect of both developmental stage and sex, we performed ANOVAs on aligned rank-transformed observations according to the general procedure outlined by *Feys, 2016* using the software R (*R Development Core Team, 2021*) and the R library ARTool (*Matthew and Wobbrock, 2020*).

To compare self-selected protein and carbohydrate consumption rates of lab-reared to field-collected fifth- and sixth-instar nymphs, we used a Mann–Whitney $U$ test (non-normal distribution). Oxygen consumption was measured in sixth-instar nymphs. We compared resting mass-specific oxygen consumption between field and lab sixth-instar nymphs using Mann–Whitney $U$ tests (non-normal distribution).

The assess the scaling of protein consumption across animals, we accounted for phylogenetic struc-ture in the trait data, using Phylogenetic Generalized Least Squares (PGLS). First, a time-calibrated phylogenetic tree with 15,029 leaf nodes was obtained from TimeTree.org (*Kumar et al., 2022*) with the following query: Schistocerca, Teleostei, *Rattus rattus*, *Gallus gallus*, *Felis catus*, *Sus domesticus*, *Rangifer tarandus,* and *Bos taurus*. 'Teleostei' was the most sensible representative taxon for the various fish species from *Dabrowski, 1986*, which spanned from salmonids to cyprinids. Using the ape package in R (*R Development Core Team, 2024*), this tree was pruned down to the focal taxon

names with the exception that '*Oncorhynchus mykiss*', one of the species analyzed in *Dabrowski, 1986*, was substituted for 'Teleostei' since there was no leaf node corresponding to the latter in the original tree. This substitution is expected to have minimal impact on the analysis given the arbitrary nature of selecting a specific teleost species to represent this deep evolutionary lineage with only a single representative in our pruned dataset. The pruned, rooted tree is depicted in *Figure 3—figure supplement 1*. Using the caper package (*Orme et al., 2023*), we integrated the phylogenetic information with trait data using the 'comparative.data' function. Subsequently, the PGLS model was fitted using the 'pgls' function.

## Acknowledgements

We thank Kelly O'Meara and Geoffrey Osgood, our former lab technicians, for helping with organization and logistics. Aunmolpreet Chahal assisted with experimental setup, as well as provided the locusts with diets and water tubes. We thank Craig Perl for assisting us with statistical analysis of phylogeny correction for protein consumption across animal species. Special thanks to Mai and Tom Talal for cleaning hundreds of experimental cages, water tubes, and diet dishes, when experiments were finished. We also thank out South American collaborators for helping with collecting the colony and helping with field logistics: Eduardo V Trumper (INTA, Argentina), Luis Sanchez Shimura (SENASAG, Bolivia), Fernando Copa Bazán (Universidad Autónoma Gabriel René Moreno, Bolivia), Jorge Frana (INTA, Argentina), and Julio E Rojas (SENAVE, Paraguay). In the US, the authors recognize that the ASU campus community has and continues to benefit from land that was taken from Indigenous communities, including the Akimel O'odham (Pima) and Pee Posh (Maricopa) Indian Communities, whose stewardship of these lands allows us to be here today. This work was supported by NSF IOS-1826848 and BARD FI-575-2018 grants.

## Additional information

### Funding

| Funder | Grant reference number | Author |
| --- | --- | --- |
| National Science Foundation | NSF IOS-1826848 | Arianne J Cease |
| US-Israel Binational Agricultural Research and Development Fund | BARD FI-575-2018 | Stav Talal |

The funders had no role in study design, data collection and interpretation, or the decision to submit the work for publication.

### Author contributions

Stav Talal, Conceptualization, Data curation, Formal analysis, Investigation, Visualization, Methodology, Writing – original draft, Writing – review and editing; Jon F Harrison, Arianne J Cease, Conceptualization, Resources, Supervision, Funding acquisition, Investigation, Methodology, Project administration, Writing – review and editing; Ruth Farington, Jacob P Youngblood, Hector E Medina, Rick Overson, Investigation, Methodology, Writing – original draft, Writing – review and editing

### Author ORCIDs

Stav Talal [iD] https://orcid.org/0000-0003-1181-5291
Jon F Harrison [iD] https://orcid.org/0000-0001-5223-216X

### Decision letter and Author response

Decision letter https://doi.org/10.7554/eLife.88933.sa1
Author response https://doi.org/10.7554/eLife.88933.sa2

## Additional files

### Supplementary files
• MDAR checklist

### Data availability
All data generated or analyzed during this study are included in the manuscript and supporting files; source data files have been provided for *Figures 1–3*.

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
