## [Editor Report]

How and why nutritional requirements change over development and differ between species are important questions with wide-ranging implications across a range of disciplines, from ecology to health. In this important study, Talal and colleagues set out to address these questions in laboratory and field experiments with grasshoppers, and with comparative analyses across different species. The laboratory experiments are convincing, and the study offers evidence of a universal shift from high protein to high carbohydrate intake during ontogeny.

---

## [Decision Letter]

**Decision letter after peer review:**

Thank you for submitting your article "Body mass and growth rates predict protein intake across animals" for consideration by *eLife*. Your article has been reviewed by three peer reviewers, one of whom is a member of our Board of Reviewing Editors, and the evaluation has been overseen by Christian Rutz as the Senior Editor.

The reviewers have discussed their reviews with one another, and the Reviewing Editor has drafted this decision letter to help you prepare a revised submission.

Essential Revisions:

Your findings reveal a predicted shift from high protein:carbohydrate consumption to lower protein:carbohydrate intake from the first instar to adulthood in many species – a decline that strongly correlated with a decrease in mass-specific growth rate. Although the reviewers find your work impressive, there are fundamental issues that need to be addressed. Below, we summarise the strengths and weaknesses of your study and provide some recommendations for revision based on the combined assessment of the reviewers; this is followed by more detailed comments from the individual reviews.

Strengths:

(1) Your study compares behaviour/physiology of laboratory vs. wild locusts. Captivity can change behaviour and physiology of most organisms, making it difficult to establish the relevance of laboratory experiments to what happens in the real world. We like your comparison between field- and laboratory-raised locusts, but this requires some corrections for specific differences between field and laboratory conditions.

(2) The anticipation that the observed trend in S. cancellata will extend to all animals based on the expectation that growth scales hypometrically across various body sizes and developmental stages adds weight and novelty to your hypothesis.

(3) Your application of the Geometric Framework (GF) of nutrition, which is a powerful approach for studying effects of nutrition and understanding the rules of compromise associated with balancing dietary unbalances.

(4) The further step of proposing a new scaling rule based on your study's results and data from the literature on various species.

Weaknesses:

(1) The implication of the 'new scaling rule key' as determined from your hypothesis test was not made clear. This lack of clarity is reflected in the apparent lack of depth in the questions outlined in lines 358-363.

(2) The assumption that field-collected animals have experienced higher stress levels (line 124) needs to be substantiated. Captivity can constitute stress depending on what is being considered. The laboratory locust results were compared independently with field-collected data for late instar nymphs of the same locust species, and the conclusion is drawn that field insects ingested similar protein but 50-90% more carbohydrate (with only 23% increased mass-specific resting oxygen consumption rates). However, numerous uncontrolled variables between the lab and field studies make meaningful conclusions difficult to draw from this observation.

(3) In the laboratory experiment, an obvious omission is tests of whether locusts did indeed feed non-randomly and converged on a common bi-coordinate intake point under the two separate food-pairing treatments. It appears that you are not estimating "Intake Targets", as stated throughout the manuscript. According to the geometric framework, the intake target (IT) is estimated as the point in the nutritional landscape under which performance/fitness is optimized. The geometric framework also predicts that animals can reach their intake targets by feeding selectivity when given a choice of diets that differ in nutrient amounts, which is what you did here. However, because the relationship between fitness/performance with diet was not established, in the choice experiments authors seem to be assuming (but not testing) that locusts are reaching their intake target. These methodological issues obscure the interpretation/conclusions presented in the manuscript.

(4) The data on late-instar field collected locusts do not address the core point of the paper about changes with development, and seem problematic in several ways, e.g., only late instar insects are tested; the previous nutritional history of insects is unknown; rearing temperatures and presumably light regimen were different; insects were not laboratory-adapted so perhaps more stressed when confined; data are not presented across successive days to indicate whether, e.g., the higher carbohydrate intakes reflected redressing of a previously incurred energy deficits early on in the stadium, or whether the differences were sustained across the stadium.

(5) The comparison between mass-specific protein consumption and specific growth rate may be problematic, as both variables seem to be estimated using final mass. You estimated a mass-specific protein intake for each instar. It is not clear why mass-specific intake and not just intake of protein was used for analysis. While the mass (or size) of an individual may influence food consumption, it seems like the authors calculated mass-specific consumption using each instar's final mass, which would make mass a result of protein consumption (and not the opposite).

Some specific comments:

Line 177 – developmental time?

Line 194 – remove name from parenthesis.

Line 265 – Protein, but not carbohydrate?

Lines 348 – 350 – I think it should be captive animals do not need to travel long distances to forage.

Panel B: correct the typo 'Flield females'

Lines 72-73 – foods, not diets. Diets comprise one or more foods (in this case, two).

Lines 79-82. Not entirely poorly understood – worth adding a bit more here. Other experimental and comparative data and theoretical modeling on sources of interspecific variation in protein to non-protein intake targets that could have been mentioned include changes across trophic levels, with phylogenetically independent acquisition of nitrogen-upgrading endosymbionts, and with adaptation to features of the nutritional environment (e.g., cats vs domesticated dogs, domesticated dogs vs wolves). There is some literature on each of these topics, which could be cited.

Line 89. About milk and early development, I don't think any mammal has a protein-biased intake target in the sense of P: nP >1. Human milk is ~7%P of total energy, for example.

Lines 123-125. Seems a bit ad hoc as a prediction – as above.

Lines 160-163. Not mentioned again in the manuscript – did they defend an intake target?

Lines 183-186. Some questions that occurred to me: Temperature and light regimens therefore differed from the lab – was a source of radiant heat provided in the field as in the lab? How did P: C change across the 8 days, relative to changes across days in the lab? Was there a big spike in carbohydrate intake during the early days, indicative of redressing an earlier food shortage? Were insects placed on diets at moulting as in the lab or at a less defined age during the stadium, which would skew towards overall greater C intake if C intake is low early in the stadium?

Line 265. Typo in the subheading.

Lines 304-305. Yes, only partially.

(Lines 47-49) "While a few studies indicate developmental effects on macronutrient intake, we lack a clear understanding about how and why ontogeny and body size affect consumption and intake targets (Peters, 1983)." Perhaps some more recent papers have contributed to this topic. For example see: Ojeda-Avial et al. 2003 doi.org/10.1016/S0022-1910(03)00003-9; Wang et al. 2019 doi: 10.1093/jisesa/iez098

I think that the paper would benefit from clarifying the differences between ontogeny and body size. Although both are correlated, these are not the same, especially in animals with discontinuous growth, like insects. In some sections, it seems that ontogeny and body size are used interchangeably. In the title, for example, was body mass or ontogeny what predicted changes in protein intake?

Line 135 states that animals were excluded if dying. Could authors clarify if the death of animals was random or higher in a specific diet treatment?

Line 145-147 the writing in this section is a bit hard to understand.

Line 182 Can the fact that wild-collected locusts were from an outbreak influence their intake targets and physiology in general?

Some of the methods sections are a bit confusing (e.g. Lines 145-148)

Line 135 "…experiments, Animals…" animals should be lowercase.

Eq. correct type, should be "developmental"

If this was already presented in methods, then do not repeat in results (Lines 220-221; 241-243)

Recommendation for revision

(1) The main goal of this study was to test how and why the intake of two important macronutrients ‒protein and carbon‒ often changes with ontogeny and body size. The laboratory experiments, in which different instars have been provided with one of two nutritionally complementary food pairings differing in protein to carbohydrate (P:C) content, and their self-selected protein to carbohydrate "intake target" measured provide an elegant dataset to address this.

(2) Although the protein: carbohydrate intake in the lab population appeared to be consistent with that observed in a wild locust population, I do not think the two should be compared directly. A more robust approach is the come up with a prediction or a set of predictions based on the laboratory experiment and test it with field data. This may require intake rate and body mass measurements from several instars.

(3) Finally, a more formal meta-analysis/comparative biology approach is required before the data in Figure 3 are convincing. How were cases for inclusion sourced (literature search terms), what was the justification for the temperature correction to 37oC, should there be consideration of phylogenetic effects, etc.? These questions need to be addressed for the graph provided in Figure 3 showing comparative data across a selection of species which makes the case that protein consumption scales similarly both developmentally and across taxa to be convincing. This section also needs clearer justification and predictions (and any expected correction factors) based on the ecology of the selected species from the start.

*Reviewer #1 (Recommendations for the authors):*

Congratulations on a very well-written manuscript. Your findings, indeed reveal the predicted shift from high protein: carbohydrate consumption to lower protein: carbohydrate intake from the first instar to adulthood – a decline that strongly correlated with a decrease in mass-specific growth rate. I really like your comparison between field- and laboratory-raised locusts, which showed that protein demand does not differ between the field and the laboratory population, but carbohydrate consumption rate was >50% higher in the field locusts likely because of their higher activity.

What really adds further weight and novelty to your hypothesis is the anticipation that the observed trend in S. cancellata will extend to all animals based on the expectation that growth scales hypometrically across various body sizes and developmental stages. However, my primary criticism of the manuscript is that the implication of this 'new scaling rule key' determined from your hypothesis test has not been made clear. The lack of clarity is reflected in the apparent lack of depth in the questions outlined in lines 358 – 363. A stimulating question for me would be where this scaling rule does not apply, or how variation in global protein availability drives niche specialization and biogeography of animal body size, growth, development, etc., and how these may be affected by climate change.

The assumption that field-collected animals have experienced higher stress levels (line 124) needs to be substantiated. Captivity can constitute stress depending on what is being considered.

*Reviewer #2 (Recommendations for the authors):*

Regarding the lab experiment, the one omission is tests of whether locusts did indeed feed non-randomly and converged on a common bi-coordinate intake point under the two separate food-pairing treatments.

The data on late-instar field collected locusts do not address the core point of the paper in relation to changes with development, and seem problematic in several ways, e.g., only late instar insects are tested; the previous nutritional history of insects was unknown; rearing temperatures and presumably light regimen were different; insects were not lab-adapted so perhaps more stressed when confined; data are not presented across successive days to indicate whether, e.g., the higher carbohydrate intakes reflected redressing of a previously incurred energy deficits early on in the stadium, or whether the differences were sustained across the stadium.

A more formal meta-analysis/comparative biology approach is required before the data in Figure 3 are convincing. How were cases for inclusion sourced (literature search terms), what was the justification for the temperature correction to 37oC, should there be consideration of phylogenetic effects, etc.?

---

## [Author Response]

Essential Revisions:Your findings reveal a predicted shift from high protein:carbohydrate consumption to lower protein:carbohydrate intake from the first instar to adulthood in many species – a decline that strongly correlated with a decrease in mass-specific growth rate. Although the reviewers find your work impressive, there are fundamental issues that need to be addressed. Below, we summarise the strengths and weaknesses of your study and provide some recommendations for revision based on the combined assessment of the reviewers; this is followed by more detailed comments from the individual reviews.Strengths:(1) Your study compares behaviour/physiology of laboratory vs. wild locusts. Captivity can change behaviour and physiology of most organisms, making it difficult to establish the relevance of laboratory experiments to what happens in the real world. We like your comparison between field- and laboratory-raised locusts, but this requires some corrections for specific differences between field and laboratory conditions.

We have revised our explanation of the purpose of the comparison between lab and field. Because relatively few studies have made such comparisons, our goal was simply to test whether the intake targets measured with lab-reared animals would predict intake targets for field-captured animals. In the discussion, we now describe the many possible reasons that lab and field-results may differ and note that we have not shown that the scaling rules apply in the field.

(2) The anticipation that the observed trend in S. cancellata will extend to all animals based on the expectation that growth scales hypometrically across various body sizes and developmental stages adds weight and novelty to your hypothesis.

Thank you.

(3) Your application of the Geometric Framework (GF) of nutrition, which is a powerful approach for studying effects of nutrition and understanding the rules of compromise associated with balancing dietary unbalances.(4) The further step of proposing a new scaling rule based on your study's results and data from the literature on various species.Weaknesses:(1) The implication of the 'new scaling rule key' as determined from your hypothesis test was not made clear. This lack of clarity is reflected in the apparent lack of depth in the questions outlined in lines 358-363.

Thank you for the suggestion. We have added material to the discussion regarding how our discovery of this rule may affect our understanding of an organism’s physiology and ecological niche.

(2) The assumption that field-collected animals have experienced higher stress levels (line 124) needs to be substantiated. Captivity can constitute stress depending on what is being considered. The laboratory locust results were compared independently with field-collected data for late instar nymphs of the same locust species, and the conclusion is drawn that field insects ingested similar protein but 50-90% more carbohydrate (with only 23% increased mass-specific resting oxygen consumption rates). However, numerous uncontrolled variables between the lab and field studies make meaningful conclusions difficult to draw from this observation.

These are good points. We deleted the sentence indicating that field animals likely experienced higher stress levels. We also clarified the purpose of our experiments with field-collected locusts, which was to test whether our intake targets measured for lab-reared locusts could predict those of field-collected locusts; a comparison that has been rarely made. We added material to the discussion to explain some of the many possible explanations for why the field and lab populations differed in intake targets. We also clarified that the fact that metabolic rates were only 23% higher while carbohydrate consumption rates were 50-90% higher in field locusts might be due to the fact that our metabolic rates were measured on resting animals (feeding is well-known to elevate metabolic rates in locusts).

(3) In the laboratory experiment, an obvious omission is tests of whether locusts did indeed feed non-randomly and converged on a common bi-coordinate intake point under the two separate food-pairing treatments. It appears that you are not estimating "Intake Targets", as stated throughout the manuscript. According to the geometric framework, the intake target (IT) is estimated as the point in the nutritional landscape under which performance/fitness is optimized. The geometric framework also predicts that animals can reach their intake targets by feeding selectivity when given a choice of diets that differ in nutrient amounts, which is what you did here. However, because the relationship between fitness/performance with diet was not established, in the choice experiments authors seem to be assuming (but not testing) that locusts are reaching their intake target. These methodological issues obscure the interpretation/conclusions presented in the manuscript.

As defined by Raubenheimer and Simpson (2018), “the intake target (IT) is a geometric representation of the nutrient mixture that the regulatory systems target through foraging and feeding”. We did the standard procedure for measuring a self-selected intake target, using the statistical approach that is well-established in the GF literature, as documented in the references below. To exclude random feeding patterns, we carried out choice experiments for all developmental stages using two different pairs of diets that would result in locusts achieving different p:c (protein to carbohydrate) ratios if they simply ate randomly during the experiment. The insignificant effects of diet pairs (table 1 – MANCOVA output) confirm that locusts did indeed feed non-randomly and converged on a common bi-coordinate intake point under the two separate food-pairing treatments. We updated the text to make this point clear.

It is correct that we did not determine whether the intake targets we measured coincide with optimal performance. This is a complex issue because the fitness consequences of larval diet can be measured in many ways, including growth and survival of the larvae, and/or adult reproduction and adult longevity; these traits do not always correlate (Sentinella et al. 2013; Runnagal-McNaull et al. 2013). Therefore, we felt that measuring the fitness consequences of the varied intake targets for each developmental stage was beyond the scope of this paper.

We have added much of this material and the references to the discussion.

Clissold, Fiona J, Helena Kertesz, Amelia M. Saul, Julia L. Sheehan, and Stephen J. Simpson. “Regulation of Water and Macronutrients by the Australian Plague Locust, Chortoicetes Terminifera.” Journal of Insect Physiology, Mechanisms of Nutritional Homeostasis in Insects, 69 (October 1, 2014): 35–40. https://doi.org/10.1016/j.jinsphys.2014.06.011.

Harrison, Sarah J., David Raubenheimer, Stephen J. Simpson, Jean-Guy J. Godin, and Susan M. Bertram. “Towards a Synthesis of Frameworks in Nutritional Ecology: Interacting Effects of Protein, Carbohydrate and Phosphorus on Field Cricket Fitness.” Proceedings of the Royal Society B: Biological Sciences 281, no. 1792 (October 7, 2014): 20140539. https://doi.org/10.1098/rspb.2014.0539.

Raubenheimer, D., and S.J. Simpson. 2018. Nutritional ecology and foraging theory. Current Opinion in Insect Science 27: 38-45.

Runagall-McNaull, A., R. Bonduriansky and A.J. Crean. 2015. Dietary protein and lifespan across the metamorphic boundary: protein-restricted larvae develop into short-lived adults. Scientific Reports 5: 11783.

Sentinella, A.T., A.J. Crean and R. Bonduriansky. 2013. Dietary protein mediates a trade-off between larval survival and the development of male secondary sexual traits. Functional Ecology 27: 1134-1144.

Tessnow, Ashley E., Spencer T. Behmer, and Gregory A. Sword. “Protein‐carbohydrate Regulation and Nutritionally‐mediated Responses to Bt Are Affected by Caterpillar Population History.” Pest Management Science, July 29, 2020, ps.6022. https://doi.org/10.1002/ps.6022.

(4) The data on late-instar field collected locusts do not address the core point of the paper about changes with development, and seem problematic in several ways, e.g., only late instar insects are tested; the previous nutritional history of insects is unknown; rearing temperatures and presumably light regimen were different; insects were not laboratory-adapted so perhaps more stressed when confined; data are not presented across successive days to indicate whether, e.g., the higher carbohydrate intakes reflected redressing of a previously incurred energy deficits early on in the stadium, or whether the differences were sustained across the stadium.

Our goal was not to determine what specific aspect was determining the difference between lab and field animals, but to determine if lab-reared and field-caught populations had the same self-selected targets. As we mentioned above, with these uncontrolled conditions, we demonstrated that the two different developmental stages (5^th^ and 6^th^ nymphal instars) of field-collected locusts had similar protein consumption but strikingly different carbohydrate consumption rates. Comparisons of field experiments and lab data are extremely rare, especially with locust outbreaks which occur infrequently in remote areas (it had been 60 years since the last South American locust upsurge!). Therefore, while it is indeed impossible to know or record the previous history of wild marching locusts without manipulating them, we believe these comparisons are invaluable in contextualizing lab-based studies.

(5) The comparison between mass-specific protein consumption and specific growth rate may be problematic, as both variables seem to be estimated using final mass. You estimated a mass-specific protein intake for each instar. It is not clear why mass-specific intake and not just intake of protein was used for analysis. While the mass (or size) of an individual may influence food consumption, it seems like the authors calculated mass-specific consumption using each instar's final mass, which would make mass a result of protein consumption (and not the opposite).

Thank you for this important suggestion. We agree that it was problematic to use final mass when assessing the relationship between protein consumption and growth rate as final mass was an important parameter for calculation of both. Therefore, in our new analysis, when comparing protein and carbohydrate consumption to specific growth rate, we corrected consumption rates for initial mass at the relevant instar rather than final mass. Because we only measured initial masses on a subset of animals to reduce animal stress, our sample sizes were lower for this portion of the study, but the results were very similar.

Some specific comments:Line 177 – developmental time?

Thanks for catching this spelling error, which we have fixed.

Line 194 – remove name from parenthesis.

The parentheses around this reference were removed as requested.

Line 265 – Protein, but not carbohydrate?

We revised this header for clarity.

Lines 348 – 350 – I think it should be captive animals do not need to travel long distances to forage.

We agree this is clearer wording and changed the text as requested.

Panel B: correct the typo 'Flield females'

Thank you. We fixed this typo.

Lines 72-73 – foods, not diets. Diets comprise one or more foods (in this case, two).

Thanks for the correction. We fixed it in the text.

Lines 79-82. Not entirely poorly understood – worth adding a bit more here. Other experimental and comparative data and theoretical modeling on sources of interspecific variation in protein to non-protein intake targets that could have been mentioned include changes across trophic levels, with phylogenetically independent acquisition of nitrogen-upgrading endosymbionts, and with adaptation to features of the nutritional environment (e.g., cats vs domesticated dogs, domesticated dogs vs wolves). There is some literature on each of these topics, which could be cited.

As suggested, we added more examples and modified the text to reflect the fact that some of these patterns are understood.

Line 89. About milk and early development, I don't think any mammal has a protein-biased intake target in the sense of P: nP >1. Human milk is ~7%P of total energy, for example.

Thank you for this important point. We have modified the text to clarify that protein:non-protein ratio declines with ontogeny in many mammals.

Lines 123-125. Seems a bit ad hoc as a prediction – as above.

We deleted this prediction and clarified that the rationale for the use of field-collected locusts was to provide a partial test of whether our results with lab-reared animals can predict the responses of locusts collected from the field.

Lines 160-163. Not mentioned again in the manuscript – did they defend an intake target?

Yes, they did. This and other comments made it clear to us that we had not described this aspect of our results sufficiently. We have added further description of our results to hopefully clarify this point.

Lines 183-186. Some questions that occurred to me: Temperature and light regimens therefore differed from the lab – was a source of radiant heat provided in the field as in the lab?

When locusts were kept in groups before the actual experiments they had access to a radiant heat source, whether in the lab or the field. During the intake target measurements, neither field nor lab locusts had access to a radiant heat source. We clarified this in the methods.

How did P: C change across the 8 days, relative to changes across days in the lab? Was there a big spike in carbohydrate intake during the early days, indicative of redressing an earlier food shortage?

We did not measure daily macronutrient consumption so we cannot answer this interesting question. We did add some caveats to the discussion about this possibility.

Were insects placed on diets at moulting as in the lab or at a less defined age during the stadium, which would skew towards overall greater C intake if C intake is low early in the stadium?

In the lab, insects were placed on diets the day after they molted to a given instar, whereas in the field, animals were placed on diets the day after collection, so their age with in an instar was unknown. We have added a caveat to the discussion that this could contribute to the observed lab-field differences.

Line 265. Typo in the subheading.

This heading has been revised.

Lines 304-305. Yes, only partially.

We guess that this comment refers to the fact that the percentage change in metabolic rate is less than the percentage change in carbohydrate consumed. We address that issue in detail below.

-(Lines 47-49) "While a few studies indicate developmental effects on macronutrient intake, we lack a clear understanding about how and why ontogeny and body size affect consumption and intake targets (Peters, 1983)." Perhaps some more recent papers have contributed to this topic. For example see: Ojeda-Avial et al. 2003 doi.org/10.1016/S0022-1910(03)00003-9; Wang et al. 2019 doi: 10.1093/jisesa/iez098

Thank you for suggested references. We added them to our manuscript.

I think that the paper would benefit from clarifying the differences between ontogeny and body size. Although both are correlated, these are not the same, especially in animals with discontinuous growth, like insects. In some sections, it seems that ontogeny and body size are used interchangeably. In the title, for example, was body mass or ontogeny what predicted changes in protein intake?

This is a classic and important problem in scaling. In many, but certainly not all cases, patterns observed during ontogeny are similar to what is observed across within population or across species comparisons; many authors interpret this to indicate that ontogenetic patterns are caused by changes in size such as mass. In our study of the scaling of consumption in locusts, we cannot distinguish effects of age and size as these are highly correlated. In our cross-species comparison, in most cases, both ontogenetic and cross-species consumption rates were reasonably well-predicted by body mass, which suggests that size rather than age is the most important factor. We have added a sentence to the discussion indicating the difficulties of separating the effects of age and size in our experiments.

Line 135 states that animals were excluded if dying. Could authors clarify if the death of animals was random or higher in a specific diet treatment?

The death rate was relatively low and random between the two diets pairs in all experiments; we added this information to the results.

Line 145-147 the writing in this section is a bit hard to understand.

We updated the text, hopefully it’s clearer now.

Line 182 Can the fact that wild-collected locusts were from an outbreak influence their intake targets and physiology in general?

Both populations, lab, and field, were gregarious. However, certainly many aspects of life in the field may have affected the intake targets of these locusts including that they were outbreaking. We added some material to the discussion to acknowledge this important point.

Some of the methods sections are a bit confusing (e.g. Lines 145-148)

We updated the text, hopefully it’s clearer now.

Line 135 "…experiments, Animals…" animals should be lowercase.

Thanks, we fixed it.

Eq. correct type, should be "developmental"

Thanks, we fixed it.

If this was already presented in methods, then do not repeat in results (Lines 220-221; 241-243)

As suggested, we removed these sentences.

Recommendation for revision(1) The main goal of this study was to test how and why the intake of two important macronutrients ‒protein and carbon‒ often changes with ontogeny and body size. The laboratory experiments, in which different instars have been provided with one of two nutritionally complementary food pairings differing in protein to carbohydrate (P:C) content, and their self-selected protein to carbohydrate "intake target" measured provide an elegant dataset to address this.(2) Although the protein: carbohydrate intake in the lab population appeared to be consistent with that observed in a wild locust population, I do not think the two should be compared directly. A more robust approach is the come up with a prediction or a set of predictions based on the laboratory experiment and test it with field data. This may require intake rate and body mass measurements from several instars.

We certainly agree that collecting more ecological, nutritional, and physiological data will be needed to understand why the lab and field populations differed in their intake targets. Our goal was simply to test whether the lab results would be predictive for the field. Obviously, they were not. Because so few prior studies have made this comparison, we think that this is an important aspect to keep in the paper. However, we appreciate the reviewer’s perspective, and have added more sentences to the discussion regarding the many differences between lab and field populations that may affect intake targets.

(3) Finally, a more formal meta-analysis/comparative biology approach is required before the data in Figure 3 are convincing. How were cases for inclusion sourced (literature search terms), what was the justification for the temperature correction to 37oC, should there be consideration of phylogenetic effects, etc.? These questions need to be addressed for the graph provided in Figure 3 showing comparative data across a selection of species which makes the case that protein consumption scales similarly both developmentally and across taxa to be convincing. This section also needs clearer justification and predictions (and any expected correction factors) based on the ecology of the selected species from the start.

Thank you for these important comments. We now describe how we conducted our literature search for protein consumption rates during development of animals, including description of the search terms. Temperature is well-known to affect physiological rates in ectotherms, so we corrected locust and fish rates to 37°C using a Q10 of 2; we have added a reference to justify this approach. As requested, in our new analysis, we included effects of relatedness among species, using a Phylogenetic Generalized Least Squares (PGLS) analysis.

Reviewer #1 (Recommendations for the authors):Congratulations on a very well-written manuscript. Your findings, indeed reveal the predicted shift from high protein: carbohydrate consumption to lower protein: carbohydrate intake from the first instar to adulthood – a decline that strongly correlated with a decrease in mass-specific growth rate. I really like your comparison between field- and laboratory-raised locusts, which showed that protein demand does not differ between the field and the laboratory population, but carbohydrate consumption rate was >50% higher in the field locusts likely because of their higher activity.

Thank you for this comment and the appreciation of our comparison.

What really adds further weight and novelty to your hypothesis is the anticipation that the observed trend in S. cancellata will extend to all animals based on the expectation that growth scales hypometrically across various body sizes and developmental stages. However, my primary criticism of the manuscript is that the implication of this 'new scaling rule key' determined from your hypothesis test has not been made clear. The lack of clarity is reflected in the apparent lack of depth in the questions outlined in lines 358 – 363. A stimulating question for me would be where this scaling rule does not apply, or how variation in global protein availability drives niche specialization and biogeography of animal body size, growth, development, etc., and how these may be affected by climate change.

Thank you for this comment. We have revised the discussion, adding sentences covering the issues you correctly note.

The assumption that field-collected animals have experienced higher stress levels (line 124) needs to be substantiated. Captivity can constitute stress depending on what is being considered.

This is a good point. We have deleted this sentence.

Reviewer #2 (Recommendations for the authors):Regarding the lab experiment, the one omission is tests of whether locusts did indeed feed non-randomly and converged on a common bi-coordinate intake point under the two separate food-pairing treatments.

This comment caused us to realize that we had not clearly explained that we excluded random feeding by conducting choice experiments using two complementary diet pairs, and using MANCOVA to confirm that there was no effect of diet pairs (for each developmental stage). This is the standard method to identify an intake target. We have added some additional sentences to the text which will hopefully make this clear.

The data on late-instar field collected locusts do not address the core point of the paper in relation to changes with development, and seem problematic in several ways, e.g., only late instar insects are tested; the previous nutritional history of insects was unknown; rearing temperatures and presumably light regimen were different; insects were not lab-adapted so perhaps more stressed when confined; data are not presented across successive days to indicate whether, e.g., the higher carbohydrate intakes reflected redressing of a previously incurred energy deficits early on in the stadium, or whether the differences were sustained across the stadium.

As noted above, we have added sentences to the discussion to provide further explanations as to why the field-collected locusts may have differed from the lab-collected animals.

A more formal meta-analysis/comparative biology approach is required before the data in Figure 3 are convincing. How were cases for inclusion sourced (literature search terms), what was the justification for the temperature correction to 37oC, should there be consideration of phylogenetic effects, etc.?

As we mentioned above, we have added details to the methods on the search procedures. A justification for the temperature correction, and a phylogenetic analysis.